# Bioactive Compounds from an Endophytic *Pezicula* sp. Showing Antagonistic Effects against the Ash Dieback Pathogen

**DOI:** 10.3390/biom13111632

**Published:** 2023-11-08

**Authors:** Özge Demir, Haoxuan Zeng, Barbara Schulz, Hedda Schrey, Michael Steinert, Marc Stadler, Frank Surup

**Affiliations:** 1Department Microbial Drugs, Helmholtz Centre for Infection Research, Inhoffenstrasse 7, 38124 Braunschweig, Germany; oezge.demir@helmholtz-hzi.de (Ö.D.); haoxuan.zeng@helmholtz-hzi.de (H.Z.); hedda.schrey@helmholtz-hzi.de (H.S.); marc.stadler@helmholtz-hzi.de (M.S.); 2Institute of Microbiology, Technical University of Braunschweig, Spielmannstraße 7, 38106 Braunschweig, Germany; b.schulz@tu-bs.de (B.S.); m.steinert@tu-bs.de (M.S.)

**Keywords:** *Hymenoscyphus fraxineus*, biocontrol, endophytic fungi, structure elucidation, semi-synthesis, secondary metabolites, natural products, biofilm inhibition

## Abstract

A fungal endophyte originating from the Canary Islands was identified as a potent antagonist against the fungal phytopathogen *Hymenoscyphus fraxineus*, which causes the devastating ash dieback disease. This endophyte was tentatively identified as *Pezicula* cf. *ericae*, using molecular barcoding. Isolation of secondary metabolites by preparative high-performance liquid chromatography (HPLC) yielded the known compounds CJ-17,572 (**1**), mycorrhizin A (**3**) and cryptosporioptides A–C (**4**–**6**), besides a new *N*-acetylated dihydroxyphenylalanin derivative **2**, named peziculastatin. Planar structures were elucidated by NMR and HRMS data, while the relative stereochemistry of **2** was assigned by H,H and C,H coupling constants. The assignment of the unknown stereochemistry of CJ-17,572 (**1**) was hampered by the broadening of NMR signals. Nevertheless, after semisynthetic conversion of **1** into its methyl derivatives **7** and **8**, presumably preventing tautomeric effects, the relative configuration could be assigned, whereas comparison of ECD data to those of related compounds determined the absolute configuration. Metabolites **1** and **3** showed significant antifungal effects in vitro against *H. fraxineus*. Furthermore, **4**–**6** exhibited significant dispersive effects on preformed biofilms of *S. aureus* at concentrations up to 2 µg/mL, while the biofilm formation of *C. albicans* was also inhibited. Thus, cryptosporioptides might constitute a potential source for the development of novel antibiofilm agents.

## 1. Introduction

Ash dieback is a lethal disease of ash trees in Europe. It is caused by an invasive ascomycete, *Hymenoscyphus fraxineus*, introduced from Far East Asia [1]. Since its arrival in Europe, the very existence of the common ash (*Fraxinus excelsior*) is endangered. Infections with *H. fraxineus* result in deterioration of the quality of the wood, mortality of the trees and major financial losses for forest enterprises. Additionally, loss of *F. excelsior* also means loss of biodiversity, namely the fauna and microorganisms associated with the tree.

The application of antagonistic endophytes as biocontrol agents is an attractive potential choice for the management of this plant disease. Endophytic symbionts encompass all organisms, mostly bacteria and fungi, which live within healthy plant tissues without causing apparent harm to their host. Endophytic fungi are already being successfully applied as biocontrol agents, e.g., for white spruce trees [2].

The endophytic mycobiome of ash has been recently investigated by several research groups and tested in co-culture for inhibition of *H. fraxineus.* Kosewang et al. hypothesized that fungal endophytic communities of tolerant ash species protect the trees against ash dieback and that selected specific endophytes have thus potential as biocontrol agents [3]. Becker et al. [4] isolated endophytes from compound leaves of *F. excelsior*, Kowalski and Bilanski [5] from petioles in leaf litter of ash trees from the previous year and Bilanski [6] from petioles of diseased ash. These research groups tested their isolates in co-culture with *H. fraxineus*, identifying multiple inhibitory isolates which could potentially be developed into biocontrol agents. Nawrot-Chorabik et al. [7] went one step further in that they also tested two endophytes, *Thielavia basicola* and *Minimidochium* sp., in in vitro plant-fungus systems (callus cultures and callus-regenerated seedlings). When these in vitro systems were inoculated with *H. fraxineus*, they only remained healthy and showed no defense responses when in co-culture with endophytes. Additionally, the endophytes produced novel metabolites in those co-cultures. The protective effects of endophytes were also studied by Barta et al. [8], who found that the length of necroses in the trunks of ash trees formed by *H. fraxineus* was reduced when co-inoculated with endophytes. Inoculation of axenically cultured ash seedlings with endophytes was reported by Halecker et al. [9], who determined that *H. fraxineus* and some endophytes, e.g., latent pathogens, can cause disease symptoms in the seedlings. Others, such as *Hypoxylon rubiginosum*, colonized the seedlings asymptomatically. Since *H. rubiginosum* produces phomopsidin, which is toxic for *H. fraxineus*, it is a candidate for biocontrol of *F. excelsior.*

Secondary metabolites play a key role in ash dieback and associated microorganisms. The pathogen *H. fraxineus* produces viridiol, demethoxyviridiol, hyfraxinic and lactones with phytotoxic activities [10]. In addition, *H. fraxineus* produces triterpenes with cytotoxic and the decalinoyl tetramic acid hymenosetin with antibacterial activities [11,12]. These might target competing endophytic fungi and bacteria.

Although endophytes have huge potential as biocontrol agents, basic knowledge regarding their secondary metabolism is needed before they can be used for biocontrol. Herein, we describe the finding of *Pezicula* cf. *ericae* 8999 as an active agent against *H. fraxineus* in co-culture and the comprehensive investigation of its secondary metabolites.

## 2. Materials and Methods

### 2.1. General Spectroscopic Instrumentation

Optical rotations were recorded in methanol and in ethanol (Uvasol, Merck, Darmstadt, Germany) using an MCP-150 polarimeter (Anton-Paar Opto Tec GmbH, Seelze, Germany) at 20 °C. Electronic circular dichroism (ECD) spectra were measured using a Jasco J-815 spectropolarimeter (JASCO, Pfungstadt, Germany).

Nuclear magnetic resonance (NMR) spectra were recorded on an Avance III 500 spectrometer with a BBFO (Plus) SmartProbe (Bruker, Billerica, MA, USA, ^1^H-NMR: 500 MHz and ^13^C-NMR: 125 MHz) and an Ascend 700 spectrometer with 5 mm TCI cryoprobe (Bruker, Billerica, MA, USA, ^1^H-NMR: 700 MHz and ^13^C-NMR: 175 MHz).

High-resolution electrospray ionization mass spectrometry (HR-ESIMS) spectra were acquired with an Agilent 1200 Infinity Series HPLC-UV system (Agilent Technologies, Santa Clara, CA, USA) utilizing a C18 Acquity UPLC BEH (waters) column (2.1 × 50 mm, 1.7 µm), solvent A: water +0.1% formic acid, solvent B: acetonitrile + 0.1% formic acid, gradient: 5% B for 0.5 min increasing to 100% B in 19.5 min and then maintaining 100% B for 5 min, flow rate 0.6 mL/min and UV/Vis detection 200–640 nm connected to a MaXis ESI-TOF mass spectrometer (Bruker, Billerica, MA, USA) (scan range 100–2500 *m*/*z*, capillary voltage 4500 V, dry temperature 200 °C).

Electrospray ionization mass (ESIMS) spectra were recorded using an UltiMate 3000 Series uHPLC (Thermo Fischer Scientific, Waltman, MA, USA; column 2.1 × 50 mm, 1.7 µm, C18 Acquity HPLC BEH (waters), solvent A: water + 0.1% formic acid, solvent B: acetonitrile + 0.1% formic acid, gradient: 5% B for 0.5 min increasing to 100% B in 19.5 min, then isocratic condition at 100% B for 5 min, a flow rate of 0.6 mL/min and diode array detection (DAD) of 210 nm and 190–600 nm) connected to an amaZon speed ESI-Iontrap-MS (Bruker, Billerica, MA, USA).

### 2.2. Fungal Material

The endophyte strain *Pezicula* cf. *ericae* 8999 (DSM 110620) was isolated from the shoot tissues of the shrub *Viburnum tinus* from Gomera Island (Canary Islands, Spain) by B. Schulz and Siegfried Draeger in May 2006 and tentatively identified as *Cryptosporiopsis* sp., probably owing to the fact that the culture showed a cryptosporiopsis-like anamorph, which is characteristic of the genus *Pezicula* in the current sense [13]. Meanwhile, *Cryptosporiopis* has actually been merged into *Pezicula* according to the 1F1N concept (Index Fungorum current name: *Pezicula ericae* (Sigler) P.R. Johnst) [14]. However, the genus *Pezicula* is a member of the Dermateaceae in the ascomycete order Helotiales, which is in bad need of a taxonomic revision. For instance, the type of *Cryptosporiopsis ericae* was obtained from huckleberry (Ericaceae) in Canada, while the host of strain *Pezicula* cf. *ericae* 8999 was reported as a species of *Viburnum* (Adoxaceae) and no morphological comparison with the type strain has been carried out. Given the current state of knowledge about the diversity of Ascomycota, as well as the knowledge of the biogeography of the Canary Islands, it appears rather improbable that the same fungus can occur in the Macaronesian Archipelago as well as in the forests of Northwestern America.

The comparison of ITS and LSU rDNA sequences with those deposited in Gen-Bank at least confirmed the placement of the sequences derived from strain Pezicula cf. ericae 8999 in the genus Pezicula and suggested placement within the species ericae. However, ITS and LSU sequences were recently demonstrated to be unsuitable for the identification of fungal species, because of the high degree of polymorphisms that were detected in the genomes on the one hand and the high redundancy of rDNA sequences within particular genera on the other hand [15]. Therefore, we refrain from giving this strain a species name and prefer to refer to it tentatively by the name Pezicula cf. ericae.

The strain is deposited at the Leibniz Institute DSMZ (German Collection of Microorganisms and Cell Cultures GmbH) with the designation no. DSM 110620 and further taxonomic studies to establish its true identity are presently pending. Sequence data was deposited under GenBank accession no. OR755921.

### 2.3. Fermentation and Extraction

*Pezicula* cf. *ericae* 8999 was grown on YM6.3 (Yeast–Malt medium; 10 g/L malt extract, 4 g/L d-glucose, 4 g/L yeast extract, 20 g/L agar, pH 6.3) agar plates for 7 days at 23 °C; then, three small pieces of well-grown mycelium from the agar plates were transferred into two 250 mL Erlenmeyer flasks, each containing 100 mL of yeast–malt extract broth (10 g/L malt extract, 4 g/L d-glucose, 4 g/L yeast extract, pH 6.3). The seed cultures were incubated at 23 °C on a rotary shaker at 140 rpm. After 5 days of cultivation, an Ultra-Turrax (T25 easy clean digital, IKA), equipped with an S 25 N–25 F dispersing tool was used to homogenize the culture broth.

For the solid rice medium, 10 mL of seed culture was transferred into two 500 mL Erlenmeyer flasks, containing solid rice-based medium [100 mL of the base liquid (1 g/L yeast extract, 0.5 g/L sodium tartrate, 0.5 g/L K_2_HPO_4_) added to 28 g of brown rice (“Biograde” Langkorn Naturreis procured from a local supermarket, Kaufland)] and incubated for 30 days at 23 °C. Firstly, the surface of the mycelium was covered with acetone and it was sonicated in an ultrasonic bath for 40 min at 39 °C. The mycelium was separated from the acetone using paper filters. This procedure was repeated twice and both acetone extracts were combined, followed by evaporation with a rotary evaporator. The remaining aqueous residue was extracted with the same amount of ethyl acetate twice. The organic phase was evaporated to dryness in vacuo at 39 °C.

For the liquid Potato Dextrose broth (PDB) medium, 6 mL of seed culture was transferred to each of four 500 mL Erlenmeyer flasks, containing potato–dextrose broth (PDB; HiMedia, Mumbai, India), and incubated for 5 days at 23 °C on a rotary shaker at 140 rpm. The consumption of the glucose was monitored daily using glucose test stripes (Medi-Test Glucose, Macherey–Nagel, Düren, Germany).

The fermentation was completed 3 days after the glucose depletion. The supernatant and the mycelium were separated via vacuum filtration. The mycelium was extracted with acetone three times in an ultrasonic water bath (Sonorex Digital 10 P, Bandelin Electronic GmbH & Co. KG, Berlin, Germany) at 39 °C for 40 min. The extracts were combined and the solvent evaporated. The remaining aqueous phase was diluted with distilled water and extracted with the same amount of ethyl acetate three times. Subsequently, the organic phase was evaporated to dryness in vacuo (39 °C). The supernatant was extracted with an equal amount of ethyl acetate twice and this organic phase was evaporated to dryness (39 °C).

This yielded 148 mg of extract from the mycelium and 191 mg of extract from the supernatant of PDB cultures, as well as 196 mg of extract from rice cultures.

### 2.4. Isolation of Metabolites

First, all the crude extracts were filtrated by using an SPME Strata-X 33 µm polymeric reversed-phase (RP) cartridge (Phenomenex, Aschaffenburg, Germany).

Liquid PDB culture: The supernatant and mycelia extracts were purified separately using preparative reverse-phase HPLC (PLC 2250, Limburg, Germany). The extract obtained from the mycelium of the culture was purified using the VP Nucleodur 100-5 C18ec column (250 × 21 mm, 7 µm: Machery–Nagel, Düren, Germany), solvent A: Deionized water (Milli-Q, Millipore, Schwalbach, Germany) with 0.1% formic acid and solvent B: acetonitrile with 0.1% formic acid. The flow rate was set to 20 mL/min and UV absorption was measured at 210, 254 and 350. The elution gradient was 50% solvent B for 3 min, 50–80% B for 50 min, 80–100% B for 10 min and then maintained at 100% B for 10 min. This yielded compound **1** (5.5 mg, t_R_ = 13.2 min). The supernatant extract was purified using the same equipment and elution conditions as mentioned. This yielded compound **2** (3.9 mg, t_R_ = 13.2 min) and compound **3** (2.7 mg, t_R_ = 7.4 min).

Solid rice culture: The extract obtained from the solid rice culture was purified using preparative reverse phase HPLC (PLC 2250, Limburg, Germany). A VP Nucleodur column 100-5 C18ec column (250 × 21 mm, 7 µm: Machery–Nagel, Düren, Germany) was used as the stationary phase. Deionized water (Milli-Q, Millipore, Schwalbach, Germany) with 0.1% formic acid (solvent A) and acetonitrile with 0.1% formic acid (solvent B) were used as the mobile phase. The flow rate was set to 20 mL/min and the elution gradient was 45% solvent B for 3 min, 45–80% B for 40 min, 80–100% B for 20 min and then maintained at 100% B for 10 min. This yielded compound **4** (4.3 mg, t_R_ = 11.8 min), compound **5** (3.6 mg, t_R_ = 12.6 min) and compound **6** (2.2 mg, t_R_ = 13.3 min).

### 2.5. Spectral Data

CJ-17,572 (**1**): white powder, [α]^20^_D_ = +109 (c = 0.1, MeOH); ^13^C NMR (500 MHz, CHCl_3_-*d*): *δ*_C_ 200.4, 192.5, 177.0, 129.3, 129.2, 100.6, 66.7, 66.5, 48.6, 42.3, 39.0, 38.6, 35.7, 35.6, 33.5, 28.3, 27.2, 22.5, 18.4, 17.0, 14.3 ppm; HR-ESIMS: *m/z* 362.2326 [M + H]^+^ (calculated for C_21_H_32_NO_4_, 362.2326); data are in good agreement with those of [16].

Compound **2**: colorless oil; [α]^20^_D_ = −153 (c = 0.1, MeOH) UV (MeOH, c = 0.01 mg/mL) *λ*_max_ (log ε) 298 (4.5), 198 (4.5) nm; ^1^H NMR (500 MHz, CH_3_OH-*d*_4_): *δ*_H_ 7.37 (dd, *J* = 7.7, 1.4 Hz, 9–H), 7.04 (td, *J* = 7.7, 1.4 Hz, 7–H), 7.01 (m, 3′–H), 6.77 (td, *J* = 7.7, 0.9 Hz, 8–H), 6.71 (dd, *J* = 7.7, 0.9 Hz, 6–H), 6.48 (dd, *J* = 14.9, 10.7 Hz, 5′–H), 6.23 (dd, *J* = 14.9, 11.2 Hz, 4′–H), 6.12 (dd, *J* = 15.1, 10.7 Hz, 6′–H), 6.01 (d, *J* = 15.1 Hz, 2′–H), 5.76 (dd, *J* = 15.1, 8.2 Hz, 7′–H), 5.65 (d, *J* = 2.6 Hz, 3–H), 5.04 (d, *J* = 2.6 Hz, 2–H), 2.20 (m, 8′–H), 1.22–1.36 (m, 9′–H_2_, 10′–H_2_, 11′–H_2_, 12′–H_2_, 13′–H_2_), 1.01 (d, *J* = 6.7 Hz, 15′–H_3_), 0.90 (t, *J* = 7.0 Hz, 14′–H_3_) ppm; ^13^C NMR (125 MHz, CH_3_OH-*d*_4_): *δ*_C_ 174.5 (C, C–1), 169.1 (C, C–1′), 155.3 (C, C–5), 146.4 (CH, C–7′), 142.5 (CH, C–3′) 141.6 (CH, C–5′), 129.9 (CH, C–6′), 129.6 (CH, C–4′), 129.3 (CH, C–7), 128.8 (C, C–4), 128.2 (CH, C–9), 123.7 (CH, C–2′), 120.1 (CH, C–8), 115.8 (CH, C–6), 70.3 (CH, C–3), 58.1 (CH, C–2), 38.6 (CH, C–8′), 38.2 (CH_2_, C–9′), 33.2 (CH_2_, C–12′), 30.7 (CH_2_, C–11′), 28.6 (CH_2_, C–10′), 23.9 (CH_2_, C–13′), 21.0 (CH_3_, C–15′), 14.6 (CH_3_, C–14′); HR-ESIMS: *m/z* 416.2434 [M + H]^+^ (calculated for C_24_H_34_NO_5_, 416.2431).

Mycorrhizin A (**3**): yellowish [α]^20^_D_ = +24 (c = 0.1, EtOH); ^1^H NMR (500 MHz, CH_3_OH-*d*_4_): *δ*_H_ 6.98 (s, 3–H), 6.94 (q, *J* = 6.9 Hz, 2′–H), 2.26 (dd, *J* = 8.2, 5.8 Hz, 9–H), 2.02 (d, *J* = 6.9 Hz, 3′–H_3_), 1.78 (dd, *J* = 8.2, 4.6 Hz, 10–H_a_), 1.78 (dd, *J* = 8.2, 4.6 Hz, 10–H_a_), 1.78 (dd, *J* = 5.8, 4.6 Hz, 10–H_b_), 1.31 (s, 11–H_3_), 1.21 (s, 12–H_3_); ^13^C NMR (125 MHz, CH_3_OH-*d*_4_): *δ*_C_ 194.4 (C, C–2), 193.9 (C, C–5), 147.4 (C, C–4), 138.0 (CH. C–2′), 135.9 (CH, C–3), 128.5 (C, C–1′), 103.0 (C, C–6), 83.2 (C, C–8), 45.3 (CH, C–9), 44.3 (C, C–1), 29.3 (CH_3_, C–12), 25.5 (CH_3_, C–11), 16.4 (CH_3_, C–3′), 15.5 (CH_2_, C–12); HR-ESIMS: *m/z*: 283.0734 [M + H]^+^ (calculated for C_14_H_16_ClO_4_, 283.0732), data are in good agreement with the literature [17].

Cryptosporioptide A (**4**): yellow powder; HR-ESIMS: *m/z* 751.1503 [M + H]^+^ (calculated for C_36_H_31_O_18_, 751.1505); data are in good agreement with the literature [18].

Cryptosporioptide B (**5**): yellow powder; HR-ESIMS: *m/z* 779.1817 [M + H]^+^ (calculated for C_38_H_35_O_18_, 779.1818); data are in good agreement with the literature [18].

Cryptosporioptide C (**6**): yellow powder; HR-ESIMS: *m/z* 807.2131 [M + H]^+^ (calculated for C_40_H_39_O_18_, 807.2131); data are in good agreement with the literature [18].

### 2.6. Methylation of CJ-17,572 with Diazomethane

Diazomethane in diethyl ether was prepared using an Aldrich diazomethane generator. A total of 3 mL of ether was added to the outside tube of the apparatus. To the inside tube, 0.367 g diazald and carbitol (1 mL) were added. The lower part of the outer tube was immersed in an ice bath. A total of 1.5 mL of 37% aqueous potassium hydroxide (KOH) was slowly injected through the rubber septum via a syringe. The apparatus was gently shaken to ensure the mixing of reactants within the inner tube. Diazomethane was produced in the inner tube as a result of the reaction between alkali and diazald, escaped via a vent, and then was collected in the outer tube in diethyl ether.

Subsequently, 7.1 mg of **1** were dissolved in 2 mL MeOH and 2 mL distilled water was added. When the diazomethane was ready to use, 2 mL ether/diazomethane solution was added. After 90 min of stirring, the solvent was evaporated under a nitrogen dryer.

Purification was performed by RP HPLC (Gilson PLC 2050) using a Nucleodur C18 ec column (250 × 10 mm, 5 µm), solvent A: Milli-Q H_2_O + 0.1% formic acid, solvent B: acetonitrile + 0.1% formic acid, flow rate: 5 mL/min, gradient: 3 min B at 60%, increasing to 80% in 15 min and increasing to 100% in 3 min followed by isocratic conditions with 100% for 3 min. Fraction 1 led to 2.2 mg compound **7** and fraction 2 led to 1.8 mg compound **8**.

4′-methyl-CJ-17,572 (**7**): white crystals [α]^20^_D_ = +93 (c = 0.1, MeOH); UV (MeOH, c = 0.05 mg/mL) *λ*_max_ (log ε) 282 (3.8), 234 (3.8), 201 (3.8) nm; ^1^H NMR (500 MHz, CHCl_3_-*d*): *δ*_H_ 5.56 (dd, *J* = 9.8, 4.6 Hz, 4–H), 5.32 (d, *J* = 9.8 Hz, 5–H), 4.16 (dqd, *J* = 10.5, 6.6, 3.1 Hz, 6′–H), 3.91 (d, *J* = 3.1 Hz, 5′–H), 3.84 (s, 4′–OCH_3_), 3.19 (qd, *J* = 6.8, 4.6 Hz, 3–H), 2.92 (s, 1′–CH_3_), 2.06 (d, *J* = 10.5 Hz, 6′–OH), 1.77 (m, 6–H_a_), 1.732 (m, 9a–H), 1.731 (m, 5a–H), 1.69 (m, 8–H_a_), 1.45 (m, 9–H_a_), 1.43 (m, 7–H), 1.27 (s, 2′–CH_3_), 1.19 (d, *J* = 6.6 Hz, 7′–H_3_), 1.07 (m, 8–H_b_), 0.906 (m, 9–H_b_), 0.904 (m, 3′–CH_3_), 0.899 (m, 3′–CH_3_), 0.85 (m, 6–H_b_); ^13^C NMR (125 MHz, CHCl_3_-*d*): *δ*_C_ 206.1 (C, C–1), 171.6 (C, C–4′), 169.6 (C, C–2′), 130.8 (CH, C–4), 128.6 (CH, C–5), 114.9 (C, C–3′), 66.7 (CH, C–6′), 66.4 (CH, C–5′), 62.5 (CH_3_, 4′–OCH_3_), 53.7 (C, C–2), 42.1 (CH_2_, C–6), 39.4 (CH, C–9a), 38.3 (CH, C–5a), 37.8 (CH, C–3), 35.6 (CH_2_, C–8), 33.0 (CH, C–7), 28.6 (CH_3_, 1′–CH_3_), 26.5 (CH_2_, C–9), 22.4 (CH_3_, 7–CH_3_), 19.0 (CH_3_, 3–CH_3_), 18.4 (CH_3_, C–7′), 17.6 (CH_3_, 2–CH_3_) ppm; HR-ESIMS: *m/z* 376.2487 [M + H]^+^ (calculated for C_22_H_34_NO_4_, 376.2482).

2′-methyl-CJ-17,572 (**8**): white crystals [α]^20^_D_ = +13 (c = 0.1, MeOH); UV (MeOH, c = 0.03 mg/mL) *λ*_max_ (log ε) 286 (4.1), 201 (3.9) nm; ^1^H NMR (500 MHz, CHCl_3_-*d*): *δ*_H_ 5.55 (ddd, *J* = 9.8, 4.8, 2.2 Hz, 4–H), 5.33 (br d, *J* = 9.8 Hz, 5–H), 4.26 (d, *J* = 10.7 Hz, 6′–OH), 4.16 (dqd, *J* = 10.7, 6.4, 4.9 Hz, 6′–H), 4.05 (s, 2′–OCH_3_), 3.70 (d, *J* = 4.9 Hz, 5′–H), 3.32 (m, 3–H), 3.00 (s, 1′–CH_3_), 1.77 (m, 6–H_a_), 1.76 (m, 5a–H), 1.69 (m, 9a–H), 1.68 (m, 8–H_a_), 1.49 (m, 9–H_a_), 1.45 (m, 7–H), 1.36 (s, 2–CH_3_), 1.12 (m, 8–H_b_), 1.10 (d, *J* = 6.4 Hz, 7′–H_3_), 0.94 (m, 9–H_b_), 0.90 (d, *J* = 6.4 Hz, 7–CH_3_), 0.85 (m, 6–H_b_), 0.71 (d, *J* = 7.2 Hz, 3–CH_3_); ^13^C NMR (125 MHz, CHCl_3_-*d*): *δ*_C_ 201.5 (C, C–1), 191.9 (C, C–4′), 180.5 (C, C–2′), 130.7 (CH, C–5), 128.9 (CH, C–4), 103.6 (C, C–3′), 67.9 (CH, C–5′), 66.6 (CH, C–6′), 63.4 (CH_3_, 2′–OCH_3_), 52.2 (C, C–2), 42.4 (CH_2_, C–6), 39.5 (CH, C–9a), 38.0 (CH, C–5a), 35.7 (CH_2_, C–8), 34.2 (CH, C–3), 33.2 (CH, C–7), 30.0 (CH_3_, 1′–CH_3_), 27.3 (CH_2_, C–9), 22.5 (CH_3_, 7–CH_3_), 18.9 (CH_3_, 3–CH_3_), 17.2 (CH_3_, C–7′), 15.9 (CH_3_, 2–CH_3_) ppm; HR-ESIMS: *m/z* 376.2484 [M + H]^+^ (calculated for C_22_H_34_NO_4_, 376.2482).

### 2.7. Degradation and Marfey’s Analysis of CJ-17,572

Compound **1** (2 mg) was treated with MeOH (25 mL) and sodium hypochlorite (NaOCl) (1.5 mL). The reaction mixture was stirred at room temperature for 20 min and the mixture was evaporated to dryness. The residue was dissolved in water (10 mL) and washed with chloroform (CHCl_3_) (10 mL) twice. The mixture was separated using a separation funnel. Subsequently, the water phase was hydrolyzed with 6n hydrochloric acid (HCl) (2 mL) at 110 °C for 15 h in a sealed vial. The hydrolysate was dried under a nitrogen dryer and redissolved in Milli-Q water (200 µL). Then, 1n sodium bicarbonate (NaHCO_3_) (20 µL) was added and the mixture was divided into two portions. Half of the hydrolyzed product was derivatized with 1% N-(2,4-dinitro-5-fluorophenyl)-D-valinamide (D-FDVA, 100 µL in acetone) and the other half was derivatized with 1% L-FDVA (100 µL in acetone). Each mixture was heated at 40 °C for 40 min. After cooling, the solutions were neutralized with 2N HCl, and the samples were dried.

The amino acid found in compound 1 was used as the standard. The standard amino acid N-methyl threonine was separately derivatized with d-FDVA and l-FDVA under the same procedure as that used for Marfey’s analysis of compound 1.

Subsequently, all the resulting products were dissolved in 1 mL MeOH and analyzed using a HPLC system connected to an amaZon speed ESI- Iontrap mass spectrometer.

### 2.8. Antimicrobial Assay

The antimicrobial activity (minimum inhibitory concentration, MIC) of the compounds was determined against *Acinetobacter baumannii* DSM 30008, *Bacillus subtilis* DSM 10, *Chromobacterium violaceum* DSM 30191, *Escherichia coli* DSM 1116, *Mycobacterium smegmatis* ATCC 700084, *Pseudomonas aeruginosa* PA14, *Staphylococcus aureus* DSM 346, *Candida albicans* DSM1665, *Mucor hiemalis* DSM 2656, *Pichia anomala* DSM 6766, *Rhodotorula glutinis* DSM 10134 and *Schizosaccharomyces pombe* DSM 70572 in a serial dilution assay as previously described [19].

### 2.9. Cytotoxicity Assay

The in vitro cytotoxicity assay was performed using the MTT (3-(4,5-dimethylthiayol-2-yl)-2,5-diphenyltetrazolium bromide) test in 96-well microtiter plates as previously described [19].

### 2.10. Agar Diffusion Assay

The antifungal activity of the isolated compounds against *H. fraxineus* was assessed by the disc diffusion method using 9 cm diameter petri dishes as previously described [9] (by using two-layered agar-based Potato–Dextrose medium (potato extract dextrose agar; Carl Roth GmbH & Co. KG, Karlsruhe, Germany): a solid lower layer (20 mL of medium containing 2% agar, pH: 5.6) and a soft upper layer (6 mL of medium containing 1% agar, pH: 5.6). An amont of 20, 50 and 100 µg of the compounds were applied onto the filter paper discs. Firstly, all the isolated compounds except methylation products (**7** and **8**) were tested with 50 µg/disc. The ones which showed activity were further tested with 20 µg/disc and the ones which did not show any activity with 50 µg/disc were tested with 100 µg/disc. Three sterile filter paper discs per plate were gently placed on dried potato–dextrose agar. The fungicide nystatin was applied as the positive control (20 and 50 µg/paper disc) and methanol as the negative control (10 µL/paper disc). The test plates were incubated at 21 °C for 7 days, the inhibition diameters were measured, and the activity was evaluated compared to nystatin.

### 2.11. Biofilm Assay

*Staphylococcus aureus* DSM 1104 from −20 °C stock was cultured in 25 mL CASO (Casein–Peptone Soymeal–Peptone; peptone from casein 15 g/L, peptone from soy flour 5 g/L, NaCl 5 g/L, pH 7.3) medium at 37 °C at 100 rpm in 250 mL for 20 h. The precultured suspension of *S. aureus* was adjusted so that its OD_600_ matched the turbidity of a 0.001 McFarland standard. *S. aureus* was incubated in 96-well tissue microtiter plates (TPP tissue culture ref.no 92196, Switzerland) for 24 h at 150 rpm in 150 mL CASO medium with 4% glucose broth. The supernatant was removed from the wells and 150 μL of the respective media (fresh) was added to the wells, together with the serially diluted compounds (250–2 μg/mL). The plates were incubated for a further 24 h at 37 °C. Staining of the biofilms was carried out as described previously [19]. Standard deviation (SD) of two repeats with duplicates each were 10% or less. Methanol (2.5%) and microporenic acid A (250–2 μg/mL) were used as solvent and positive controls. Error bars indicate SD with duplicates in two biological repeats.

The stock of fungal pathogen *Candida albicans* DSM 11225 from −20 °C was cultured in 25 mL YPD (Yeast extract Peptone Dextrose; dextrose 20 g/L, peptone 20 g/L, yeast extract 10 g/L) medium in a 250 mL flask at 30 °C with shaking 100 rpm for 18 h. Turbidity of the broth was measured and diluted to match the turbidity of a 0.05 McFarland standard in RPMI 1640 medium (Gibco, New York, NY, USA, Thermo Fisher Scientific). Afterwards, 150 μL of the fungal dispersion was added into each well of a 96-well microtiter plate (Falcon no. 351172, Thermo Fisher Scientific) and further incubated for 90 min. The supernatant was removed, and the plate was washed once by using PBS buffer. Compounds were serially diluted in 150 μL in fresh medium to concentrations of 250–2 μg/mL. Staining and analysis were conducted as previously described [20]. Methanol (2.5%) and farnesol (250–2 μg/mL) were used as solvent and positive controls. Error bars indicate SD with duplicates in two biological repeats.

*Pseudomonas aeruginosa* (PA 14) DSM 19882 from −20 °C stock was precultured in 25 mL LB medium (Luria–Bertani broth; tryptone 10 g/L, NaCl 10 g/L, yeast extract 5 g/L, pH 7.0) with a 250 mL flask at 37 °C with shaking 100 rpm overnight. The turbidity of the broth was measured and diluted to match the turbidity of a 0.1 McFarland standard in M63 medium, which is supplemented with magnesium sulphate, glucose and casamino acids [20]. The compounds were added into 150 μL bacterial solution (concentration range 250–2 μg/mL), after which the solution was added to U-bottom 96-well microtiter plates (Falcon ref.no 351177, Thermo Fisher Scientific). The plates were incubated at 37 °C at 150 rpm for 24 h and biofilms were established at the air liquid interface. Staining of the biofilms and analysis was conducted as described previously [19]. Methanol (2.5%) and myxovalargin A (250–2 μg/mL) were used as the solvent control and positive control, respectively. The assay was conducted once.

Differences between samples and the control group were determined by a two-tailed Student’s *t*-test. Statistical significance was defined as *p* < 0.01. Analysis was carried out using GraphPad Prism 9^®^ (Version Prism 9.0.0, GraphPad Software, San Diego, CA, USA).

### 2.12. Phytotoxicity Assay

The phytotoxicity of the compounds **1**, **3** and **4** was assessed using the leaf puncture assay as described previously [21,22]. The other compounds could not be evaluated because of the limited quantity.

For this assay, similar leaves of the same *Fraxineus excelsior* plant were chosen onto which the compounds were applied. The pure compounds were dissolved at 1 mg/mL in MeOH. Then, 20 µL of test samples were applied to the adaxial sides of leaves that had previously been needle punctured. A total of 20 µL of MeOH in distilled water (4%) was applied on the leaves as the negative control and 1 mg/mL macrocidin A [23] as the positive control. Each treatment was repeated three times and then the leaves were placed on moistened paper filters in petri dishes (diameter 9 cm) to prevent the droplets from drying. The experiment was run in duplicate. Leaves were evaluated for the symptoms after 3 days.

## 3. Results

### 3.1. Initial Experiments

Strain *Pezicula* cf. *ericae* 8999 became a candidate for isolation and elucidation of secondary metabolites due to its excellent inhibition of *H. fraxineus* in co-culture, with a 39% growth inhibition of *H. fraxineus*. At this point, only the strains 10395 and 10403 showed stronger inhibitory effects in our screening campaign [9]. Thus, strain *Pezicula* cf. *ericae* 8999 was selected for an in-depth investigation of its secondary metabolism.

The strain was cultivated on six different solid media (potato–dextrose agar medium (PDA), yeast–malt medium (YM 6.3), cottonseed malt medium (Q6 ½), sugar malt medium (ZM ½), biomalt medium (BM) and rice medium) and five different liquid media (PDB, YM6.3, Q6 ½, ZM ½ and BM), resulting in highly diverse secondary metabolite profiles (Appendix A). In most of the studied media, **4** was produced as the main metabolite, while **1** was produced most prominently in YM and ZM media. Additionally, extracts from cultivation on the rice medium and in the liquid PDB medium showed prominent activity (2.3 µg/mL) against *Bacillus subtilis*. Hence, main metabolites **1**–**6** (Figure 1) were isolated by preparative HPLC.

### 3.2. Structure Elucidation

The molecular formula of **1** was identified as C_21_H_31_NO_4_ by its *m/z* molecular ion cluster at *m/z* 361.2326 in the HR-ESIMS spectrum. Peaks in the ^1^H and ^13^C NMR spectra were broad, though some signals were even missing. A literature search with SciFinder^N^ and Dictionary on Natural Products yielded the known 3-decanoyl tetramic acids CJ-17,572 [16] and cryptocin [24,25] as possible explanations, of which **1** was identified as CJ-17,572 by ^13^C NMR data. Since CJ-17,572 (**1**) had been published without stereochemical information, we addressed its elucidation with a series of experiments.

Optical rotation ([α] + 109) and ECD spectrum (+2.0_279_) of **1** were similar to phomasetin [α] +94 and ECD +5.2_290_) and opposite in sign to those of equisetin ([α] −280 and ECD −8.9_290_) [26], indicating a 2R absolute configuration. Since the assignment of C–5′ from ECD data appeared too ambiguous, we degraded **1** with NaOCl and subsequent HCl treatment [27]. This process released N-Me-Thr, whose configuration was determined as l- by Marfey’s method [28].

Tetramic acids are prone to tautomeric effects. For 3-acyltetramic acid, four tautomers, involving two sets of rapidly interconverting internal tautomers are normally detected in the solution. Arising from C–C bond rotation of the acyl side chain, the interchange of two pairs of external tautomers is slow on the NMR time scale, causing the broadening of signals [29]. To prevent this tautomerism, we acetylated **1** by treatment with acetic anhydride. Unfortunately, even under cooling conditions at 0 °C, an elimination reaction took place, leading to the loss of stereochemistry at C–6′. However, methylation with diazomethane yielded two main mono-methylation products (Figure 1) with molecular formulae of C_22_H_33_NO_4_, which were subsequently isolated by preparative HPLC. HMBC correlations (Appendix A) confirmed the structures of **7** and **8** with methylation of C–2′ and C–4′, respectively. Signals in ^1^H and ^13^C NMR spectra of **7** and **8** were sharp, confirming tautomerism was suspended by methylation. Subsequently, stereochemistry was addressed by a detailed analysis of **7**: ROESY correlations and ^13^C chemical shifts were highly similar to those of hymenosetin, confirming that the decanoyl moieties are identical concerning the relative configuration. Analogous to hymenosetin, the 5′S,6′R stereochemistry was determined by a J-resolved analysis of **7**.

Taken together, the 2S,3S,5aR,7S,9aS,5′S,6′R configuration was determined for **1**. The very same structure including stereochemistry was recently published under the name 5′-epi-cryptocin [25], demonstrating CJ-17,572 and 5′-epi-cryptocin are identical. Since the name CJ-17,572 was coined first, we retain CJ-17,572 for **1**.

Compound **2** was isolated as a colorless oil by preparative HPLC. Its quasimolecular ion cluster at *m/z* 416.2434 in the HRESIMS spectrum suggested the molecular formula C_24_H_33_NO_5_. ^1^H and HSQC spectra of **2** revealed the presence of two methyls, five methylenes and ten olefinic/aromatic methines as well as three other methines. COSY and TOCSY data assembled the spin systems 2–H/3–H, 6–H/7–H/8–H/9–H and 2′–H to 15′–H_3_ (see Appendix A). HMBC correlations, most notably from 2–H to C–1/C–4/C–1′, from 3–H to C–1/C–4/C–5/C–9, from 6–H to C–4/C–5/C–8 and from 2′–H to C–1′/C–3′/C–4′ elucidated the structure of a 3,5-dihydroxyphenylalanin, N-acetylated by an 8-methyltetradeca-2,4,6-trienoyl moiety. All trans geometry of the double bonds was deduced by the large coupling constants of J_2′,3′_ = 15.1 Hz, J_4′,5′_ = 14.9 Hz and J_6′,7′_ = 15.1 Hz, respectively. A J-based configuration analysis [30], which utilizes ^2^J_C,H_ and ^3^J_C,H_ carbon–proton spin coupling constants besides proton–proton spin coupling constants (^3^J_H,H_), established the 2*S**,3*R** relative configuration (Figure 2). Therefore, the J_C,H_ coupling constants were determined by HSQC-Hecade and J-HMBC NMR experiments. Absolute stereochemistry and the configuration of C–8 in the new structure of **2** are left unassigned.

Metabolite **3** was analyzed for a molecular formula of C_14_H_15_O_4_Cl by HRESIMS analysis. ^1^H, ^13^C and 2D NMR data as well its positive optical rotation identified **3** as (+)-mycorrhizin A [17]. Last but not least, cryptosporioptides A–C (**4**–**6**) were identified by HRESIMS and NMR data, metabolites already known to be produced by strain *Pezicula* cf. *ericae* 8999 [18].

### 3.3. Biological Activities

The antifungal activities of the isolated compounds were investigated in an agar diffusion assay against the plant pathogen *H. fraxineus* (isolate RH02-T6-B4-2). CJ-17,572 (**1**) and Mycorrhizin A (**3**) showed strong inhibitory effects when 50 and 20 µg test substance were applied per paper disc, respectively (Appendix A) (Appendix A).

Furthermore, the compounds were evaluated for their antibacterial and antifungal activities against a panel of bacteria and fungi. MIC values showed that compounds **1**, **2** and **3** have moderate to weak antifungal activity (Table 1). The most significant activity was noted for CJ-17,572 (**1**) against *Bacillus subtilis* and *Mucor hiemalis* with MIC values of 8.3 µg/mL and 4.2 µg/mL, respectively. This is in agreement with the results of Sugie et al. (2002), who also reported strong activities against Staphylococci [16]. Of the cryptosporioptides (**4**–**6**), the only inhibition was that of **6** against *M. hiemalis* with a MIC of 66.6 µg/mL at the highest tested concentration. The methylation products **7** and **8** showed strongly reduced activity: while 4′-methyl-CJ-17,572 (**7**) showed weak antimicrobial activity, 2′-methyl-CJ-17,572 (**8**) did not show any activity against the tested microorganisms.

Additionally, a cytotoxicity assay was conducted against a panel of mammalian cell lines. The cytotoxicity results showed that compounds **1**, **3**, **7** and **8** were active against all the tested cell lines, whereas cryptosporioptides A–C (**4**–**6**) showed no activity against the tested cell lines KB3.1 and L929 (see Table 2). Mycorrhizin A (**3**) was the most active of the metabolites against the human breast adenocarcinoma MCF-7 cell line with half-maximal inhibitory concentrations (IC_50_) of 0.64 µM, being in accordance with data previously reported [17].

In addition to the assessment of biological activities against bacteria, fungi and selected mammalian cell lines, cryptosporiopitide A (**4**), B (**5**) and C (**6**) were further investigated for their inhibitory effects on biofilms of the pathogens *P. aeruginosa* and *C. albicans* as well as for dispersal effects on preformed biofilms of *S. aureus*. Cryptosporiopitide B (**5**) and C (**6**) effectively dispersed preformed biofilms of *S. aureus* (Figure 3) with an efficacy of ca. 70–80% of biofilm eradication at concentrations between 15.6 µg/mL–125 µg/mL. Moreover, cryptosporiopitide C (**6**) exhibited dispersal effects of 40% even up to 2 µg/mL. Interestingly, cryptosporiopitide A (**4**), which is lacking the alkylic side chain in the malonate subunit, exhibited a much weaker effect. In addition to this, all of the compounds were weakly to moderately active against biofilm formation by *C. albicans* (Appendix A), while none of them inhibited *P. aeruginosa*.

Moreover, the phytotoxic activities of CJ-17,572 (**1**), mycorrhizin A (**3**) and cryptosporioptide A (**4**) were assessed with a leaf puncture assay (Appendix A). This assay showed that all the tested compounds caused brownish lesions, of which **3** showed the most phytotoxic effects, being more prominent than the positive control.

## 4. Discussion

*Pezicula* cf. *ericae* 8999 was found to be an effective antagonist against *H. fraxineus* in co-culture experiments. Investigating the secondary metabolism, we found that *Pezicula* cf. *ericae* 8999 produces CJ 17,572 (**1**) and mycorrhizin (**3**), two strongly antifungal metabolites, which might explain the observed antagonistic activity of the strain against *H. fraxineus*. However, *Pezicula* cf. *ericae* 8999 will be unsuitable for the biocontrol of *H. fraxineus* due to its production of cytotoxic and phytotoxic metabolites.

Interestingly, *H. fraxineus* produces with hymenosetin a structurally very similar compound to **1**. The only differences are the *N*-methylation and the enantiomeric relationship between their decalinoyl moieties. Both compounds showed a broad spectrum of inhibitions including gram-positive bacteria, yeasts and filamentous fungi. The effects of the two compounds on filamentous fungi are comparable. Whereby, hymenosetin was especially active against gram-positive bacteria with MIC values of 1.0 µg/mL and below for *Micrococcus luteus*, *Mycobacterium diernhoferi*, *Nocardia* sp., *Nocardioides simplex*, *S. aureus* and the methicillin-resistant *S. aureus* (MRSA) strain N315e [11]. In contrast, **1** showed only weak antibacterial effects against *B. subtilis* and *S. aureus* with MIC of 8.3 µg/mL for both. Right now, it is not clear if the methylation or the contrary stereochemistry are responsible for this altered antibacterial activity.

In general, 3-decalinoyltetramic acids have a broad activity spectrum. Equisetin, the best investigated member of this family, was isolated from *Fusarium equisetin* and *Fusarium heterosporum* and found to display extensive biological activities including antibiotic activity, HIV inhibitory activity, cytotoxicity and mammalian DNA-binding capabilities [29]. Equisetin was also phytotoxic, suppressed germination, inhibited the growth of various monocotyledonous and dicotyledonous seeds as well as young seedlings and caused necrotic lesions on the roots, cotyledons, and coleoptiles of plant seedlings. Similarly, both **1** and hymenosetin might have phytotoxic effects.

The number and complexity of secondary metabolites produced by *Pezicula* cf. *ericae* 8999 is astonishing. Compound **2** has structural similarity to gymnostatin N, which was isolated in the course of a high throughput screen against the anti-cancer target POLO-like kinase 1 [31]. Furthermore, the structure of gymnostatin N was patented for its insecticidal activities [32]. In our experiments, **2** showed only weak antibacterial and antifungal effects, but might enhance the potential of the more active compounds **1** and **3**. Mycorrhizin A (**3**) is one of the rather uncommon examples with both enantiomers having been isolated as natural products. Strong antifungal as well as cytotoxic effects have been reported for both enantiomers [17,33]. (+)-Mycorrhizin A (**3**) was highly active against *Heterobasidion annosum*, formerly known as *Fomes annosus*, which is considered to be one of the most economically important forest pathogens in the northern hemisphere. Since (+)- mycorrhizin A (**3**), which inhibits *H. annosum*, is produced by a monotropoid fungus found on the roots of *Monotropa hypopitys*, an ecological role for **3** causing antagonism in nature is supported.

In contrast, a distinct and hitherto unknown effect on preformed biofilms of *S. aureus* was observed for the cryptosporioptides **4**–**6**, for compounds **5** and **6** even at concentrations in the low µg/mL range. Our observations suggested that enhancing lipophilicity could potentially enhance the dispersal properties of cryptosporioptides. In particular, cryptosporioptides with an increase in length and number of lipophilic side chains might have stronger effects on the eradication of preformed biofilms compared with those lacking such lipophilic side chains. This observation occurs in a concentration range below cytotoxic (IC_50_ > 37 µg/mL) and lethal effects (MIC > 66.6 µg/mL), making it a promising candidate for future therapeutic applications. According to the National Institute of Health, biofilms cause approximately 60–80% of all microbial infections and no specific treatments are presently available [34,35]. Consequently, persistent infections associated with biofilms pose continuing challenges in human health care. Biofilms are highly resistant to conventional antibiotics and thus novel therapeutic strategies are urgently needed to tackle this problem [36].

## 5. Conclusions

We isolated three additional compounds, which were not previously known from the *Pezicula* cf. *ericae* 8999, which showed in vitro activity against the ash dieback pathogen *H. fraxineus*. CJ-17-572 (**1**), whose complete stereochemistry we elucidated, and (+)-mycorrhizin A (**3**) were strongly antifungal and may explain the antifungal potential of *Pezicula* cf. *ericae* 8999. Since both compounds **1** and **3** were also markedly cytotoxic, these compounds are probably not eligible for drug discovery programs and might restrict the use of *Pezicula* cf. *ericae* 8999 as a biological agent as well. In contrast, cryptosporioptides **4**–**6** showed virtually no antibiotic activity nor cytotoxicity, but a strong antibiofilm activity, and might thus constitute a potential source for the development of antibiofilm agents. Additionally, follow-up experiments might target the interaction of the various antifungal secondary metabolites in nature, which might interact synergistically.

## Data Availability

All data generated are in the manuscript or the Appendix A. Raw data (i.e., NMR and MS files) are available on request from the corresponding author.

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
