# Peer review of "Bioactive Compounds from an Endophytic Pezicula sp. Showing Antagonistic Effects against the Ash Dieback Pathogen"

_biomolecules, 2023, doi:10.3390/biom13111632_

Round 1

Reviewer 1 Report

Comments and Suggestions for Authors

 1. In this research authors tried to explore the potentials of bioactive compounds of endophyte  Pezicula sp. against DBM and other pathogens which is the need of the current time to explore sustainable solutions to prevent plant disease and improve crop production. Authors explored the possibility of such compounds and identified them in the manuscript 

2. Such type of research is necessary to investigate novel compounds due to increasing resistance in the pests against chemical pesticides 

3. This manuscript will increase the list of new bioactive compounds that have the potential as biocontrol agents

4. In methodology, they can explain in details and need to add the reason on choosing the concentration of active compound in bioassay like antifungal activity

The manuscript can be improved as per following comments given below. 

1. The introduction needs to improve for better presentation and interest of the reader

2. Add the reason behind the role of substrate in active metabolite production which was not discussed. 

3. On what basis concentration 50microg/paper disc of extract was fixed for antifungal activity

Comments on the Quality of English Language

minor revision required 

Author Response

The manuscript can be improved as per following comments given below. 

  1. The introduction needs to improve for better presentation and interest of the reader

Reply: We slightly rephrased parts of the introduction. However, in general we assess the introduction to be clear.

  1. Add the reason behind the role of substrate in active metabolite production which was not discussed. 

Reply: Thank you for this comment! We added a short discussion about the secondary metabolite pattern produced by strain 8999 in various media to the manuscript. Additionally, we added figure S40 for a fast impression which compounds are produced in various media.

  1. On what basis concentration 50microg/paper disc of extract was fixed for antifungal activity

Reply: We chose 20µg, 50µg and 100µg, since these are typical amounts of test substance commonly used in this type of experiment.

Reviewer 2 Report

Comments and Suggestions for Authors

This manuscript describes the isolation of six metabolites from endophytic fungi Pezicula cf. Ericae (tentative), strain8999, one of then described here for the first time; for another one, a detailed stereochemical elucidation is provided. All compounds have been evaluated for their antifungal, antibacterial, cytotoxic effects, as well as their inhibitory effects on biofilms of P. aeruginosa and C. albicans and on their dispersal effects on preformed biofilms of S. aureus. This manuscript deserves publication but there are a certain number of issues which must be addressed before acceptance.

Please, include structure numbering for compounds 1,2,7 and 8; otherwise, following structural elucidation discussion becomes difficult.

Structures for phomasetin, equisetin (ref 23) and hymenosetin (ref 11) must be included to improve readability.

Diagrams including proposed preferred conformations for 2 and 7 must be included and selected ROESY correlations must be shown over those preferred conformations.

Please, indicate, at the end of the structural elucidation of compound 2, that this is a novel compound.

In lines 480 and 542, it is stated that compounds 1 and 3 are strong antifungal compounds, but this is not apparent from data on table 1. Please, clarify.

In line 482, mention is made to a very similar compound to 1. A structure (and citation) for that compound must be included.

In line 502, mention is made to gymnostatin N; please, provide structure.

In line 504, mention is made to “a structurally similar N-acylated tyrosin derivative”; please provide structure.

In lines 505-506, it is stated that “In our experiments, 2 showed only weak antibacterial and antifungal effects, but might enhance the potential of the more active compounds 1 and 3”; please provide experimental evidence for this or shed some light on the basis for this statement (cite sources, if appropriate).

In lines 512-513, statement “an ecological for 3 in natural antagonism is supported” seems incomplete; authors maybe intending to refer to ecological role; please, clarify.

Paragraph between lines 529-536 must be rewritten, as it is not very descriptive of the intended description of a wide variety of structurally diversity of metabolites in Pezicula; be more precise with citations and cite structural classes, please.

In line 543, it is stated that compounds 1 and 3 are highly cytotoxic, so this may prevent their use in drug Discovery programs; on the other hand, there is no discussion on that topic previously on the text. Please, explain why observed toxicity makes them highly toxic, as comparison is made with epothilone B, which is several orders of magnitude more toxic than 1 and 3.

Comments on the Quality of English Language

No suggestions here; please, have a look at the review above.

Author Response

Please, include structure numbering for compounds 1,2,7 and 8; otherwise, following structural elucidation discussion becomes difficult.

Reply: Atom numbers were added for compounds 1-8 to improve readability as suggested.

Structures for phomasetin, equisetin (ref 23) and hymenosetin (ref 11) must be included to improve readability.

Reply: The additional figure S42 was added to the SI for a fast comparison of CJ-17,572 to phomasetin, equisetin and hymenosetin as suggested.

Diagrams including proposed preferred conformations for 2 and 7 must be included and selected ROESY correlations must be shown over those preferred conformations.

Reply: We included the new figure S40 to depict the important COSY, HMBC and ROESY correlations used in the structure elucidation process for 2, 7 and 8.

Please, indicate, at the end of the structural elucidation of compound 2, that this is a novel compound.

Reply: We added the words “in the new structure of 2” to highlight that 2 is a novel compound as suggested.

In lines 480 and 542, it is stated that compounds 1 and 3 are strong antifungal compounds, but this is not apparent from data on table 1. Please, clarify.

Reply: Both 1 and 3 are inhibiting the growth of test organism Mucor hiemalis DSM 2656 more strongly than the positive control nystatin, so believe the term “strong inhibition” is justified.

In line 482, mention is made to a very similar compound to 1. A structure (and citation) for that compound must be included.

Reply: The structure of hymenosetin was depicted in figure S43 as described above.

In line 502, mention is made to gymnostatin N; please, provide structure.

Reply: The additional figure S41 was added to the SI for a fast comparison of peziculastatin to gymnostatin N as suggested.

In line 504, mention is made to “a structurally similar N-acylated tyrosin derivative”; please provide structure.

Reply: In fact the patent describes the structure of gymnostatin N, so we rephrase this sentence.

In lines 505-506, it is stated that “In our experiments, 2 showed only weak antibacterial and antifungal effects, but might enhance the potential of the more active compounds 1 and 3”; please provide experimental evidence for this or shed some light on the basis for this statement (cite sources, if appropriate).

Reply: It is difficult to confirm synergistic effect experimentally. Thus, we believe this to be out of scope for the current paper.

In lines 512-513, statement “an ecological for 3 in natural antagonism is supported” seems incomplete; authors maybe intending to refer to ecological role; please, clarify.

Reply: The reviewer is correct, so we rephrased this sentence.

Paragraph between lines 529-536 must be rewritten, as it is not very descriptive of the intended description of a wide variety of structurally diversity of metabolites in Pezicula; be more precise with citations and cite structural classes, please.

Reply: We agree to the reviewer that this paragraph is not descriptive. Thus, we decided to delete it.

In line 543, it is stated that compounds 1 and 3 are highly cytotoxic, so this may prevent their use in drug Discovery programs; on the other hand, there is no discussion on that topic previously on the text. Please, explain why observed toxicity makes them highly toxic, as comparison is made with epothilone B, which is several orders of magnitude more toxic than 1 and 3.

Reply: Of course epothilone B is several powers of ten more active than any of the test substances, since epothilone B is exceptionally active. However, based on the reviewer’s comment we rephrase “highly cytotoxic” to “markedly cytotoxic”.

Reviewer 3 Report

Comments and Suggestions for Authors

A paper entitled “Bioactive Compounds From an Endophytic Pezicula sp. showing antagonistic Effects Against the Ash Dieback Pathogen” by Demir et al. describes the activity of different metabolites obtained from endophytic fungal strain Pezicula sp. The authors investigate the antifungal activity against the Ash Dieback Pathogen, antibacterial activity, in-vitro cytotoxicity, and biofilm activity of obtained active compounds. In addition, the obtained compounds were characterized. The manuscript contains promising data and opens the window for the biological activity of metabolites obtained from endophytic microbes. However, the manuscript can be accepted for publication in the biomolecules journal after major revision.

1-      The abstract should contain the overall conclusion.

2-      Line 36, “Endophytic symbionts encompass all fungi and microbes..” should be “Endophytic symbionts encompass microbes..” due to fungi belonging to microbes.

3-      The introduction section should be improved to describe the diverse roles of endophytic microbes in plants and biocontrol: the following references can be helpful: https://doi.org/10.3390/biom11020140; https://doi.org/10.1007/978-981-13-8495-0_6

4-      Please add a clear hypothesis and aim of the study at the end of the introduction section.

5-      The abbreviation should be added completely the first time.

6-      Line 125, please add the components of media used. This comment will be applied to all media used in the current study.

7-      In lines 142, 362, and 363, the complete name should be mentioned at the first and mentioned after that as an abbreviation.

8-      The authors investigate the antimicrobial activity, in-vitro cytotoxicity, antifungal activity against H. fraxineus, and biofilm assay. Therefore, I recommend modifying the paper title to describe the content.

9-      Legend of Figure 1, the name of the fungal strain should be mentioned before their code (8999), please check and revise throughout the manuscript.

10-  Line 446, please cite the reference correctly. In addition, the authors separate the results and discussion sections, therefore the discussion of the obtained results should be mentioned in the discussion section.

11-  The results of biological activity are not subjected to statistical analysis, please check, and analyze it.

12-  Do the authors check antifungal activity for the CJ-17,572 (1) only? Please clarify. This comment will be applied to other biological activities.

13-  I recommend investigating biological activities especially antifungal activity for all purified compounds at different concentrations.

14-  Figure 3, please change the column from solid fill to pattern fill to distinguish in black and white copy.

15-  The conclusion should be rephrased to describe the overall conclusion and contain the limitations and prospect study.

Comments on the Quality of English Language

moderate editing of the English language required

Author Response

1-      The abstract should contain the overall conclusion.

Reply: We added a final sentence to the abstract as a conclusion as suggested.

2-      Line 36, “Endophytic symbionts encompass all fungi and microbes..” should be “Endophytic symbionts encompass microbes..” due to fungi belonging to microbes.

Reply: The term “microbes” does not include fungi. This we rephrase the sentence to “Endophytic symbionts encompass all organisms, mostly bacteria and fungi, that…”.

3-      The introduction section should be improved to describe the diverse roles of endophytic microbes in plants and biocontrol: the following references can be helpful: https://doi.org/10.3390/biom11020140; https://doi.org/10.1007/978-981-13-8495-0_6

Reply: We assess these publications to be not related to the introduction, since the first one describes growth-promoting effects while the second has plant health and abiotic stress management as its topic.

4-      Please add a clear hypothesis and aim of the study at the end of the introduction section.

Reply: Our main driver of the study was to investigate the secondary metabolism of strain 8999 in detail, as this is written in the last two sentences of the introduction section.

5-      The abbreviation should be added completely the first time.

Reply: The abbreviations were added as suggested.

6-      Line 125, please add the components of media used. This comment will be applied to all media used in the current study.

Reply: Components of the all media used were added.

7-      In lines 142, 362, and 363, the complete name should be mentioned at the first and mentioned after that as an abbreviation.

Reply: These lines were changed as suggested.

8-      The authors investigate the antimicrobial activity, in-vitro cytotoxicity, antifungal activity against H. fraxineus, and biofilm assay. Therefore, I recommend modifying the paper title to describe the content.

Reply: We have been thinking about this. However, we assess the title “Compounds with antimicrobial activity, in-vitro cytotoxicity and antifungal activity From an Endophytic Pezicula sp. showing antagonistic Effects Against the Ash Dieback Pathogen” as to complicated. Thus, we prefer to stick to “Bioactive compounds..”.

9-      Legend of Figure 1, the name of the fungal strain should be mentioned before their code (8999), please check and revise throughout the manuscript.

Reply: They were checked and revised as suggested.

10-  Line 446, please cite the reference correctly. In addition, the authors separate the results and discussion sections, therefore the discussion of the obtained results should be mentioned in the discussion section.

Reply: They were checked and revised as suggested.

11-  The results of biological activity are not subjected to statistical analysis, please check, and analyze it.

Reply: A statistical analysis with error bars etc. is not used for determining minimal inhibitory concentration assays as well as agar diffusion assays.            

12-  Do the authors check antifungal activity for the CJ-17,572 (1) only? Please clarify. This comment will be applied to other biological activities.

Reply: The antifungal activity of the compounds was investigated again. Results can be found in table 1, whereas cytotoxic activity is described in table 2.

13-  I recommend investigating biological activities especially antifungal activity for all purified compounds at different concentrations.

Reply: All purified compounds were tested firstly with 50 µg test compound, if they showed activity they were tested with 20 µg. If they did not show activity they were additionally tested with 100 µg.

14-  Figure 3, please change the column from solid fill to pattern fill to distinguish in black and white copy.

Reply: Even in black and white printouts, the columns can be distinguished easily. However, we will also be happy with a different way of presentation.

15-  The conclusion should be rephrased to describe the overall conclusion and contain the limitations and prospect study.

Reply: We added a sentence for a potential outlook. The investigation of the interaction of the various antifungal metabolites is a limitation of the study right now, but also an interesting starting point for possible follow-up experiments.

Round 2

Reviewer 3 Report

Comments and Suggestions for Authors

The manuscript is suitable for publication in the current form

Comments on the Quality of English Language

Moderate editing of English language required